# WebShop: Towards Scalable Real-World Web Interaction with Grounded Language Agents

**Shunyu Yao**[*]   **Howard Chen**[*]   **John Yang**   **Karthik Narasimhan**
Department of Computer Science, Princeton University
{shunyuy, howardchen, jy1682, karthikn}@princeton.edu

## Abstract

Existing benchmarks for grounding language in interactive environments either lack real-world linguistic elements, or prove difficult to scale up due to substantial human involvement in the collection of data or feedback signals. To bridge this gap, we develop WebShop – a simulated e-commerce website environment with $1.18$ million real-world products and $12,087$ crowd-sourced text instructions. Given a text instruction specifying a product requirement, an agent needs to navigate multiple types of webpages and issue diverse actions to find, customize, and purchase an item. WebShop provides several challenges for language grounding including understanding compositional instructions, query (re-)formulation, comprehending and acting on noisy text in webpages, and performing strategic exploration. We collect over $1,600$ human demonstrations for the task, and train and evaluate a diverse range of agents using reinforcement learning, imitation learning, and pre-trained image and language models. Our best model achieves a task success rate of $29\%$, which outperforms rule-based heuristics ($9.6\%$) but is far lower than human expert performance ($59\%$). We also analyze agent and human trajectories and ablate various model components to provide insights for developing future agents with stronger language understanding and decision making abilities. Finally, we show that agents trained on WebShop exhibit non-trivial *sim-to-real* transfer when evaluated on `amazon.com` and `ebay.com`, indicating the potential value of WebShop in developing practical web-based agents that can operate in the wild.

## 1 Introduction

Recent advances in natural language processing (NLP) and reinforcement learning (RL) have brought about several exciting developments in agents that can perform sequential decision making while making use of linguistic context [30, 50, 58]. On the other hand, large-scale language models like GPT-3 [6] and BERT [11] are excelling at traditional NLP benchmarks such as text classification, information extraction and question answering. While the former set of tasks are limited in their set of linguistic concepts and prove difficult to scale up, the latter tasks usually contain static, non-interactive datasets that lack adequate grounding to extra-linguistic concepts [4]. In order to make further progress in building **grounded** language models, we believe there is a need for **scalable** interactive environments that contain: (1) language elements that reflect rich, real-world usage and are collectible at scale, and (2) task feedback that is well-defined and automatically computable to facilitate interactive learning, without the constant need for expensive feedback from humans.

The world wide web (WWW) is a massive open-domain interactive environment that inherently satisfies the first aforementioned requirement through its interconnected set of pages with natural text, images and interactive elements. By being simultaneously **scalable, semantic, interactive, dynamic and realistic**, the web is uniquely different from existing environments for autonomous

---

[*]Equal contribution. Project site with code, data, and demos: `https://webshop-pnlp.github.io`.

36th Conference on Neural Information Processing Systems (NeurIPS 2022).

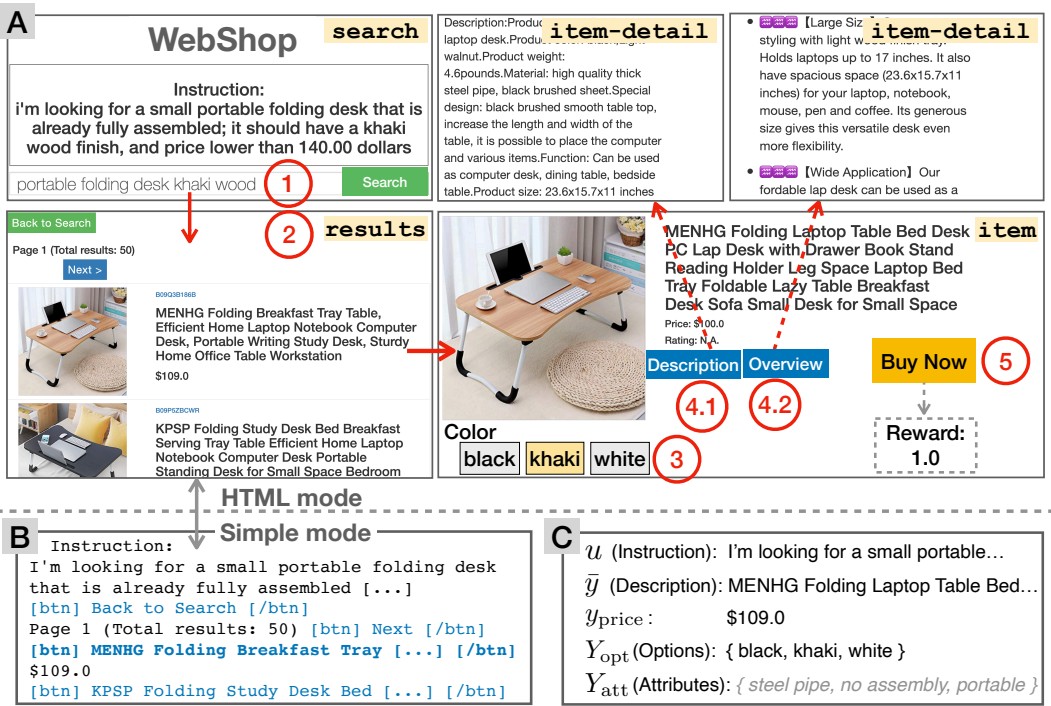

Figure 1: The WebShop environment. **A**: An example task trajectory in HTML mode, where a user can (1) search a query in a `search` page, (2) click a product item in a `results` page, (3) choose a color option in a `item` page, (4) check `item-detail` pages and go back to the item page, and (5) finally buy the product to end the episode and receive a reward $r \in [0, 1]$ (§3.2). **B**: the `results` page in `simple` mode for agent training and evaluation. The blue text indicates clickable actions and bold text indicates an action selected by the agent. **C**: The product notation used in §3 with corresponding examples from the product in **A**. The attributes $Y_{\text{att}}$ are hidden from the task performer.

agents like games or 3D navigation. Moreover, the web also provides a practical environment to deploy trained agents, with great potential for alleviating human efforts in tedious tasks (e.g. buying products, booking appointments). While there has been prior work on building web-based tasks, they either lack depth in the transition and action spaces, or prove difficult to scale up. Some benchmarks only contain either a single classification task [39, 46, 31] or interactions containing only a handful of different pages in each episode [43]. Others propose tasks with longer horizons but are either limited to following hyperlinks for web navigation [36] or require human-in-the-loop feedback due to the lack of an automated reward function [33].

In this paper, we introduce WebShop (Figure 1) – a large-scale interactive web-based environment for language understanding and decision making – and train autonomous agents to complete tasks on this benchmark. With the goals of being scalable and containing realistic language and visual elements, WebShop emulates the task of online shopping on an e-commerce website, where the agent's goal is to understand a human-provided text instruction and *purchase* a product to match the specifications. To do so, the agent needs to query the website's search engine, choose items to explore from search results, open and read their description and details, and select the necessary options (e.g. 32 oz., red color) before clicking the 'Buy' button. In order to pick the optimal product that matches user requirements, the agent may need to view and compare various products (including backtracking between pages), and potentially perform multiple searches. WebShop contains over one million products scraped from `amazon.com`, over 12 thousand crowdsourced instructions, and a diverse semantic action space of searching text queries and choosing text buttons. It is packaged into a convenient OpenAI Gym [5] environment and can be rendered in two modes (HTML or `simple`) with parallel observation spaces that are easy for human and model respectively. Rewards are automatically computed using a combination of programmatic matching functions that consider the attributes, type, options and price of the chosen product, alleviating the need for human evaluation and providing a path to scaling up interactive learning.

We develop several agents to perform this task, using both reinforcement learning (RL) and imitation learning (IL). We also leverage the latest pre-trained language models [26, 11] for representing and generating text. Our modular architecture includes a factorized processing of state observations and action choices using ResNets (visual) and Transformers (text), followed by an attention fusion layer that helps the agent contextually score each action. Our best agent achieves an average score of 62.4 (out of 100) and successfully completes the task 28.7% of the time, significantly higher than a heuristic baseline that achieves 45.6 and 9.6%, respectively. While this demonstrates the potential for IL and RL, the agents are still much lower than human experts, who can achieve 82.1 and 59.6% on this task.[*] We perform several analyses and ablation studies to identify the cause of this gap and find several avenues for agent improvement in the future including more robust search generation, explicit memory modules, and better handling of noisy web text. Finally, we also demonstrate an instance of *sim-to-real* transfer by deploying agents trained with WebShop to operate on `amazon.com` and `ebay.com`, and find that they can achieve similar performances despite search engine and product differences, and consistently outperform the rule baseline of using the first result returned by the commercial search engines when directly searching the instruction texts. This demonstrates the practical potential of our work towards developing agents that can operate autonomously on the world wide web (WWW).

## 2   Related Work

**Reinforcement learning on the web.** Nogueira and Cho [36] introduced WikiNav as a benchmark for RL agents navigating pages, but the task is purely navigational with the actions restricted to either choosing a hyperlink to follow or deciding to stop. The World of Bits (WoB) benchmark [43] enables training of RL agents to complete tasks on webpages using pixel and Document Object Model (DOM) observations. Several follow-up papers have tackled MiniWoB using techniques like workflow-guided exploration [29], curriculum and meta-learning [15], DOM tree representation [21], adversarial environment generation [16] and large-scale behavioral cloning [20]. However, MiniWoB lacks long-range decision making across multiple different pages and does not scale easily in terms of difficulty or size due to its use of low-level mouse clicks and keystrokes as actions. In contrast, WebShop requires navigating longer paths with context-based action selection and backtracking, and it uses high-level *search* and *choose* actions that are more scalable and transferable to real settings. While not directly operating on web pages, AndroidEnv [48] and MoTIF [8] provide environments to train agents for interacting with apps and services on mobile platforms.

**Non-interactive web-based tasks.** Various supervised classification tasks on webpages have been proposed, including predicting web elements [39], generating API calls [46, 47, 54] and semantic parsing into concept-level navigation actions [31]. Perhaps most similar content-wise to our work is the Klarna product page dataset [19] which contains over 50,000 product pages labeled with different element categories for supervised classification. All these works only consider supervised settings with a single decision, and may require the definition of web APIs or command templates for each domain. Our benchmark, WebShop, combines webpages with realistic text and image content with a rich and diverse interaction space for long-range sequential decision making.

**Leveraging the web for traditional NLP tasks.** Several papers have explored the use of the web for information extraction [34] and retrieval [1], question answering [57, 25], dialog [45], and training language models on webtext [2]. These approaches primarily use web search engines as a knowledge retriever for gathering additional evidence for the task at hand. Perhaps most similar to our work is WebGPT [33], which uses a web interface integrated with a search engine to train RL agents to navigate the web and answer questions. However, our environment has a more diverse action and observation space (including images) and does not require human-in-the-loop evaluation.

## 3   The WebShop Environment

We create WebShop as a large-scale web-based interactive environment with over 1.1 million real-world products scraped from amazon.com. In this environment, an agent needs to find and purchase a product according to specifications provided in a natural language instruction. WebShop is designed

---

[*]In our analysis (§5.3), we observe that the task requires patience and consistency, which is lacking in some crowdsource workers, leading to lower scores. Even with this caveat, the gap between human performance and the model remains significant.

| Type | Argument | State → Next State |
|------|----------|--------------------|
| search | [*Query*] | Search → Results |
| choose | Back to search | ∗ → Search |
| choose | Prev/Next page | Results → Results |
| choose | [*Product title*] | Results → Item |
| choose | [*Option*] | Item → Item |
| choose | Desc/Overview | Item → Item-Detail |
| choose | Previous | Item-Detail → Item |
| choose | Buy | Item → Episode End |

Table 1: Actions in WebShop.

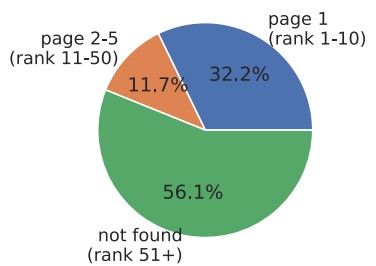

Figure 2: Item rank in search results when the instruction is directly used as search query.

in a modular fashion which disentangles the website transitions from the task-specific aspects like instructions and reward, allowing for easy extension to new tasks and domains.

### 3.1 Task Formulation

WebShop can be formulated as a partially observable Markov decision process (POMDP) $(\mathcal{S}, \mathcal{A}, \mathcal{T}, \mathcal{R}, \mathcal{U}, \mathcal{O})$ with state space $\mathcal{S}$, action space $\mathcal{A}$, deterministic transition function $\mathcal{T} : \mathcal{S} \times \mathcal{A} \to \mathcal{S}$, reward function $\mathcal{R} : \mathcal{S} \times \mathcal{A} \to [0, 1]$, instruction space $\mathcal{U}$, and a state observation space $\mathcal{O}$.

**State and action.** A state $s \in \mathcal{S}$ represents a web page, which falls into one of the four types – the *search* page that contains a search bar, the *results* page that lists a set of products returned by a search engine, the *item* page that describes a product, or the *item-detail* page that shows further information about the product (Figure 1A(1-4) respectively). We define the following notations for a product $y$. We denote $\bar{y}$ to be the aggregation of the various text fields including product title, description, and overview. We denote $y_{\text{price}}$ to be the price, $Y_{\text{opt}}$ to be a set of buying options, and $I$ to be a set of images, each corresponding to a specific option. Finally, each product is associated with $Y_{\text{att}}$, a set of attributes hidden from the agent which is extracted from the title and the *item-detail* pages (§3.2). The attributes are used for the automatic reward calculation.

An action $a \in \mathcal{A}(s)$ can either be searching a text query (e.g. search[Red shoes]) or choosing a text button (e.g. choose[Size 9]) as shown in Table 1. These two action types are not available simultaneously – search is only allowed when the agent is at the search page; on all other pages, click is the only action choice. The chosen action argument (button) will be clicked as a web link as opposed to the low-level mouse-click actions in previous environments such as World of Bits [43]. The transitions initiated by clicks deterministically redirect the web page to one of the four page types (Table 1). The transition initiated by search is based on a deterministic search engine (§3.2).

**Observation.** Using Flask [41] and OpenAI Gym [5], we provide two parallel observation modes to render the state and instruction $\mathcal{S} \times \mathcal{I} \to \mathcal{O}$: (1) HTML mode that contains the HTML of the web page, allowing for interaction in a web browser(Figure 1A), and (2) simple mode which strips away extraneous meta-data from raw HTML into a simpler format (Figure 1B). The human performance scores in §4.2 are collected in the HTML mode, while all models are trained and evaluated in the simple mode. Note that while the environment allows for training reinforcement learning agents on raw pixels in HTML mode (like in Shi et al. [43]), we believe that it provides a very low-level non-semantic action space. Moreover, it is straightforward to write a translator that converts any new HTML page into simple format for use with trained agents, which enables sim-to-real transfer.

**Instruction and reward.** Each natural language instruction $u \in \mathcal{U}$ contains the following information: a non-empty set of attributes $U_{\text{att}}$, a set of options $U_{\text{opt}}$, and a price $u_{\text{price}}$. The instruction is generated based on a target product $y^*$ by human annotators. The instruction collection process is lightweight and scalable (§3.2). Concretely, $U_{\text{att}} \subseteq Y_{\text{att}}^*$ is a subset of the product attributes, $U_{\text{opt}} \subseteq Y_{\text{opt}}^*$ is a subset of the product option field-value pairs, $u_{\text{price}} > y_{\text{price}}^*$ is a price set to be higher than the target product price. For example, the instruction "Can you find me a pair of *black-and-blue* sneaker that is *good in rain weather*? I want it to have *puffy soles*, and price less than 90 dollars." contains the aforementioned attributes $U_{\text{att}} = \{$"waterproof", "soft sole"$\}$ and option $U_{\text{opt}} = \{$"color": "black and blue"$\}$. In each episode, the agent receives a reward $r = \mathcal{R}(s_T, a)$ in the end at timestep $T$, where $a = $ choose[buy], $y$ is the product chosen by the agent in the final state $s_T$, and $Y_{\text{att}}$ and $Y_{\text{opt}}$ are its corresponding

attributes and options. The reward is defined as:

$$r = r_{\text{type}} \cdot \frac{|U_{\text{att}} \cap Y_{\text{att}}| + |U_{\text{opt}} \cap Y_{\text{opt}}| + \mathbf{1}[y_{\text{price}} \leq u_{\text{price}}]}{|U_{\text{att}}| + |U_{\text{opt}}| + 1} \tag{1}$$

where the type reward $r_{\text{type}} = \text{TextMatch}(\bar{y}, \bar{y}^*)$ is based on text matching heuristics to assign low reward when $y$ and $y^*$ have similar attributes and options but are obviously different types of products. For example, "butter" and "plant-based meat" differ in types but may both contain attributes "cruelty-free", "non-GMO", and an option "size: pack of 2". The exact formula for $\text{TextMatch}(\cdot)$ is in the Appendix §A.5.

**Evaluation metrics.** We use two evaluation metrics: (1) **Task Score**: defined as $(100 \times \text{avg. reward})$, which captures the average reward obtained across episodes; and (2) **Success Rate (SR)** defined as the portion of instructions where $r = 1$. Note that it is possible to obtain $r = 1$ for an episode even if the final product is not $y^*$ — for example, there could be many items that satisfy the goal "I want a red shirt", even if the goal is generated from a specific red shirt item.

## 3.2 Environment Implementation

**Data scraping.** We use ScraperAPI [35] to scrape $1,181,436$ products from `amazon.com` across 5 categories (fashion, makeup, electronics, furniture, and food) using 113 sub-category names as queries. The product texts (title and item details) have an average length of 262.9 and a vocabulary size $224,041$ (word frequency higher than 10). In addition, the products have a total of $842,849$ unique options, reflecting the scale and complexity of the data. More details about product scraping is in the Appendix §A.1.

**Search engine.** We use Pyserini [28] for the search engine, where indices are built offline using a BM25 sparse retriever with text for each product concatenated from the title, description, overview, and customization options. The search engine is deterministic, which eases imitation learning and result reproducibility. More details in A.3.

**Attribute mining and annotation.** Each product is annotated with a set of hidden *attributes*, which are used to represent its latent characteristics as well as to calculate the reward as detailed in §3. An attribute is a short natural language phrase that describes the property of the product (see examples in Figure 1). We mine the attributes by calculating TF-IDF scores for all bi-grams in the concatenated titles and descriptions based on each product category. We review the top 200 bi-grams for each category, remove the noisy ones by inspection (decide based on whether the bi-gram is human understandable), and assign them to the products. We consolidate a pool of 670 attributes. See more details in the Appendix §A.2.

**Natural language instructions.** We use Amazon Mechanical Turk (AMT) to collect natural language instructions that specify goal products with appropriate options. Specifically, an AMT worker is presented with a sampled goal product, including the product title, category, attributes, and the buying options, and asked to write a command to instruct an automatic shopping agent to find the target. Workers are instructed to avoid being too specific such as including the entire title in the instruction, but stay faithful to describing the target product. We collect a total of $12,087$ linguistically diverse instructions with an overall vocabulary size of $9,036$ words and an average length of 15.9 words. We provide the detailed annotation process and interface in the Appendix §A.4.

**Human demonstrations.** We collect trajectories from humans performing the task in the `HTML` mode of WebShop to understand the task difficulty for humans and to analyze how humans would solve the task. We use qualification tests to train and select motivated workers to perform the task. We recruit and train a total of 13 workers for data collection, and among them we select the top 7 performing workers to be "experts" (see Appendix §A.6 for examples). We also leverage this data to perform imitation learning (described in §4.2).

## 3.3 Research Challenges

WebShop brings together several research challenges for autonomous systems from various subfields in NLP and RL into a single benchmark. These include: 1) generation of good search queries [22, 59] and reformulation [37, 51], 2) strategic exploration for navigating through the website [55, 56, 29], 3) robust language understanding for textual state and action spaces [3, 7, 17, 44], and 4) long-term

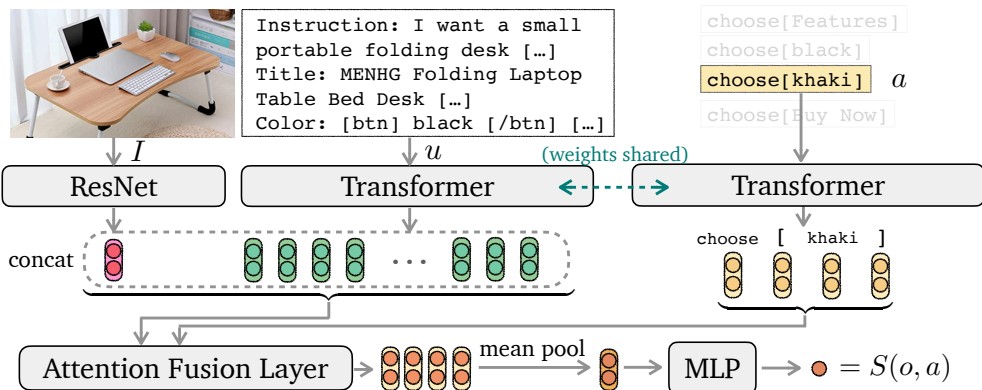

Figure 3: Architecture of our choice-based imitation learning (IL) model. The image $I$ is passed to a ResNet to obtain the image representation. The instruction text $u$ is passed to a transformer (initialized with BERT) to obtain the text representations. The concatenated bi-modal representations are fused with the action representations using the Attention Fusion Layer. The resulting fused-action representations are mean-pooled and reduced by an MLP layer to a scalar value $S(o, a)$ denoting the logit value of the action `choose[khaki]`.

memory for comparing items or backtracking [53, 13, 23] (Figure 1). While we believe individual advances in each of these will improve agent performance, WebShop also provides an ideal testbed for the development of interdisciplinary techniques that tackle more than one of the above mentioned challenges simultaneously. For example, external memory modules may be very effective if combined with strategic exploration, or exploration could be helpful in information query reformulation. Further analysis based on human and model trajectories is in §5.3.

## 4 Methods

We propose various models that combine language and image pre-training with imitation learning (IL) and reinforcement learning (RL). More details are provided in the Appendix §B.

### 4.1 Rule Baseline

A simple **rule baseline** is to search the exact instruction text, then choose and buy the first item in the results page without choosing any options. The heavy lifting of the lexical search engine makes it also a simple non-learnable information retrieval (IR) baseline, and would lead to a non-trivial attribute reward. However, simple heuristic rules cannot resolve noisy natural language options, strategically explore, or learn to generate what to search, so the total reward and task success rate should be low.

### 4.2 Imitation Learning (IL)

For the text generation and choice problems presented in WebShop, we propose using two pre-trained language models to separately learn how to search and choose from human demonstrations.

**Imitating human search generation.** We frame searching as a sequence-to-sequence text-generation problem: the agent generates a search action $a = \text{search}[\dots]$ given an instruction $u$ without considering any other context (e.g. past searches, visited items). We use $M = 1,421$ instruction-search pairs from $1,012$ training human trajectories to construct a dataset $\mathcal{D} = \{(u, a)\}_{i=1}^{M}$ and fine-tune a BART model [26] parameterized by $\phi$ to perform conditional language modeling:

$$\mathcal{L}_{\text{search}} = \mathbb{E}_{u,a \sim \mathcal{D}} \left[ -\log \pi_\phi(a \mid u) \right] \tag{2}$$

**Imitating human choice.** The choice-based imitation model (Figure 3) predicts a probability distribution over all the available click actions $\mathcal{A}(o)$ in observation $o$ and maximizes the likelihood of the human clicked button $a^* \in \mathcal{A}(o)$. We construct a dataset $\mathcal{D}' = \{(o, \mathcal{A}(o), a^*)\}_{i=1}^{M'}$ of $M' = 9,558$ samples from the training human trajectories. We use a 12-layer pre-trained BERT model [10] parameterized by $\theta$ to encode the $o$ into an observation representation of contextualized token embeddings, and we similarly encode each action. Each action representation is passed into a cross-attention layer with the observation representation, then mean pooled into a single vector

and multiplied with a matrix $W$ to obtain a scalar score $S(o, a)$. The policy $\pi_\theta\left(a \mid o, \mathcal{A}(o)\right)$ is the softmax distribution over action scores $S(o, a)$:

$$\mathcal{L}_{\text{choose}} = \mathbb{E}_{o, \mathcal{A}(o), a^* \sim \mathcal{D}'}\left[-\log \pi_\theta\left(a^* \mid o, \mathcal{A}(o)\right)\right] \tag{3}$$

$$\pi_\theta\left(a \mid o, \mathcal{A}(o)\right) \sim \exp\left(W^\top \text{mean}\left[\text{cross-attn}\left(\text{BERT}(o; \theta), \text{BERT}(a; \theta)\right)\right]\right) \tag{4}$$

**Handling Images.** We use a pre-trained ResNet-50 [18] to pre-process images across different products and options into a $512$ dimensional feature vector, which is then transformed into $768$ dimensions with a learned linear layer and concatenated to $\text{BERT}(o)$ as the observation representation.

**Full pipeline.** Combining the above during environment interaction, we use the BART model in the search page to generate the top-5 search queries via beam search and choose a random one. For other pages, we sample one action from $\pi_\theta\left(a \mid o, \mathcal{A}(o)\right)$ using the BERT model. We find these methods useful to encourage diverse actions. In contrast, an ineffective strategy that uses only the top generated search query or the button with the highest probability might lead to limited product candidates or being stuck (e.g. bouncing back and forth between pages).

### 4.3 Reinforcement Learning (RL)

We also fine-tune the choice-based IL model with online RL (i.e. IL+RL). Prior work suggests that directly fine-tuning text generation via RL might lead to language drifting [24] and deteriorated performance. Therefore, we freeze the BART model to provide the top-10 search generations as a refined action space for the choice-based IL model to learn to pick – an inspiration borrowed from previous work in text games [55] and referential games [24]. We use the policy gradient method [32] with return-to-go $R_t = \mathbb{E}_\pi[r_t + \gamma R_{t+1}]$ and a learned value baseline $V(o) = W_v^\top \text{BERT}(o; \theta)$ parameterized by $\{W_v, \theta\}$ (the BERT weights are tied with the policy):

$$\mathcal{L}_{\text{PG}} = \mathbb{E}_\pi\left[-\left(R_t - V(o_t)\right)\log \pi\left(a_t \mid o_t, \mathcal{A}(o_t)\right)\right] \tag{5}$$

The value $V(o)$ is learned with an L2 loss $\mathcal{L}_{\text{value}} = \left(R_t - V(o_t)\right)^2$. We also add an entropy loss $\mathcal{L}_{\text{entropy}} = \sum_{a \in \mathcal{A}(o_t)} \pi_\theta\left(a_t \mid o_t, \mathcal{A}(o_t)\right)\log \pi_\theta\left(a_t \mid o_t, \mathcal{A}(o_t)\right)$ to prevent premature convergence. Our full RL model minimizes the total loss $\mathcal{L}_{\text{RL}} = \mathcal{L}_{\text{PG}} + \mathcal{L}_{\text{value}} + \mathcal{L}_{\text{entropy}}$.

## 5 Experiments

### 5.1 Setup and task verification

We split a total of $12,087$ instructions into an i.i.d. distributed train / development / test split of $10,587$ / $1,000$ / $500$ instances for all models. While future work can investigate splits with more generalization gaps (e.g. split by product category), we will show the i.i.d. split is already challenging for current models. We randomly sample a subset of the $10,587$ training instructions, then collect $1,012$ human demonstrations for task verification and imitation learning (IL) and a further $54$ demonstrations from instances in the development set for IL hyperparameter tuning and checkpoint selection. We also collect human trajectories for all $500$ test instructions and report human and model performances averaged across these $500$ instructions. More setup details are in the Appendix §C.

### 5.2 Results

**Task performance.** From Figure 4, we observe that the rule baseline obtains a low score of $45.6$ and a very low success rate of $10\%$ since it cannot resolve options specified in language or explore more products, empirically demonstrating the non-trivial nature of the task. The IL model significantly outperforms the rule baseline on both metrics, achieving a score of $59.9$. Further RL finetuning improves the score to $62.4$ while slightly hurting the success rate ($29.1\% \rightarrow 28.7\%$) (analyzed further in §5.3). We also observe a significant gap between models and humans – our best model's success rate ($29.1\%$) is less than half of expert humans ($59.6\%$) and only $60\%$ of the average human ($50\%$). This indicates a great room for model improvement by tackling reseach challenges in WebShop.

**IL ablations.** Figure 4 also contains several ablations that confirm important design choices for models. When the choice action model for the IL agent is randomly initialized (**IL (w/o LP Choice)**; LP = language-pretraining), the success rate drops by nearly two-thirds, indicating the importance of language pre-training for our task. When the search query generator in the IL agent is replaced by a simple rule, which always uses the instruction text (**IL (w/o LP Search)**), both reward and success rate drop by around 3 points. This suggests the importance to explore by expanding the search space

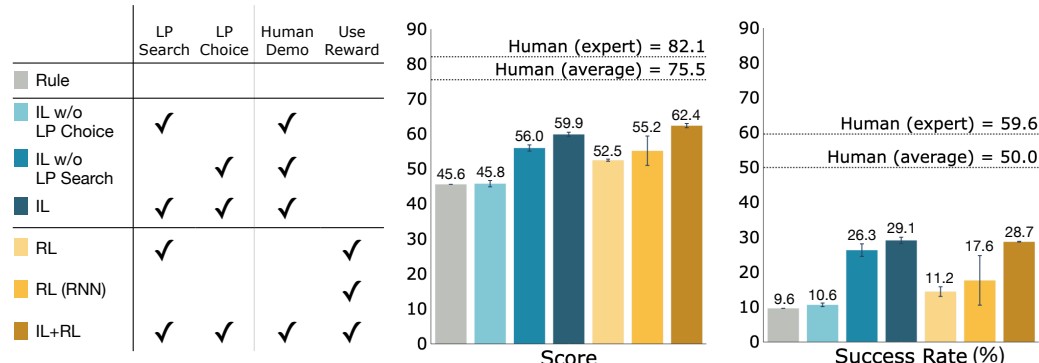

Figure 4: Task scores and Success Rate (%) for our models on the test split of WebShop over 3 trials. LP Search uses a pre-trained BART model to generate the search query and IL w/o LP Search uses the rule-based heuristic. LP Choice uses pre-trained BERT weights to initialize the choice action model and IL w/o LP Choice trains a Transformer from scratch.

| | | | Score | | | Count | | |
| | All | Att | Opt | Type | Price | State | | Item | | Search | |
|---|---|---|---|---|---|---|---|---|---|---|---|
| Rule | 45.6 | 66.6 | 0.0 | 80.5 | 86.0 | 3.0 | (3 / 3) | 1.0 (1 / 1) | 1.0 (1 / 1) |
| IL | 59.9 | 69.3 | **45.2** | 86.4 | 84.0 | 9.4 | (90 / 3) | 1.6 (11 / 1) | 1.3 (17 / 1) |
| IL+RL | **62.4** | **74.0** | 38.9 | **89.7** | **88.7** | 4.5 | (5 / 1) | 1.0 (1 / 1) | 1.0 ( 1 / 1) |
| Human Expert | 82.1 | 81.8 | 73.9 | 94.4 | 97.7 | 11.3 | (114 / 4) | 1.9 (16 / 1) | 1.4 (16 / 1) |

Table 2: Left: Score breakdown. Right: average, maximum, and minimum number of states visited, items checks, and searches in a trajectory.

for exploration, but it is not as critical as learning to choose the right options. We experiment with incorporating history of one past observation and the last five actions into the model and find a slight degradation in the score from $59.9$ to $57.3$, suggesting more advanced techniques are needed to leverage past information. More ablations in §C.

**RL ablations.** When we directly train an RL agent (**RL**) from pre-trained BERT parameters, the performance is even worse than the rule baseline. This suggests that IL warm-starting is critical, possibly because of the significant domain shift from traditional language tasks. We also consider a simple RL model with RNN text encoders instead of the Transformer (**RL (RNN)**), which has a success rate more than $10\%$ worse than the IL + RL model with a much larger variance. We hypothesize that RL with a more powerful architecture could help boost and stabilize the performance if the model is initialized with better language and task priors.

## 5.3 Analysis

To better understand the differences between the agents and human experts, we perform several fine-grained analyses. We first break down the overall score into its four sub-parts according to Eq. (1): 1) attribute score ($|U_{att} \cap Y_{att}|/|U_{att}|$), 2) option score ($|U_{opt} \cap Y_{opt}|/|U_{opt}|$), 3) price score ($\mathbf{1}[y_{price} \leq u_{price}]$), and 4) type score ($r_{type}$). We report trajectory statistics such as the average number of states, unique items visited, and number of searches per episode in Table 2 and provide qualitative examples of the trajectories in Table 3.

**Human expert vs. agents.** Human experts outperform the agents on all score sub-parts (Table 2), but the most significant boost comes from the option score (a $28\%$ gap), revealing that agents have trouble selecting the correct product options. Humans also have longer trajectories, explore more items and perform more searches than the agents, with a higher variance, demonstrating their flexibility. Table 3 provides some samples trajectories. In the first example, the human decides to search again after removing 'inches', 'width', 'height', and 'white' from the query since product texts often contain abbreviated symbols for these terms like '"', 'w', and 'h'. Thus, **search generation** is challenging for models since it involves reasoning and adapting to grounded environments, and ideas from query reformulation [37, 1] could help alleviate this. Agents also struggle to perform robust **semantic**

| Instruction 1 | Instruction 2 |
|---|---|
| I want to find white blackout shades that are 66 inches in width and 66 inches in height. They need to be easy to install [...] | I need a gingko light and 20"x20" pillow cover that is hand painted [...] |
| **Human Actions** ($r = 1.0$, length $= 8$)
search[ 66 inches in width and 66 inches in height white shades ]
choose[ item : CALYX... ]
choose[ Back to Search ]
search[ 66 x 66 blackout shades ]
choose[ item : Milin... ]
choose[ opt : 66"w x 66"h ]
choose[ opt : cordless bottom up-blackout-white ]
choose[ Buy ] | **Human Actions** ($r = 1.0$, length $= 17$)
search[ gingko light 20"x20" pillow cover hand painted ]
choose[ item : Maison... ] [...]
choose[ Description ]
choose[ < Previous ]
choose[ Overview ]
choose[ < Previous ] [...]
choose[ item : Maison... ]
choose[ opt : 20"x20" ]
choose[ opt : nudes (gingko light) ]
choose[ Buy ] |
| **IL+RL Actions** ($r = 0.2$, length $= 3$)
search[ white blackout shades 65 inches in width and 66 inches in height ]
choose[ item : Window... ]
choose[ Buy ] | **IL+RL Actions** ($r = 0.25$, length $= 3$)
search[ gingko light and 20x20 pillow cover hand painted ]
choose[ item : UPOOS... ]
choose[ Buy ] |

Table 3: Two example trajectories (showing only actions) from the human and the IL+RL model. We omit some human actions from instruction 2 for space and truncate the item names for readability. Red denotes options and blue denotes attributes.

|  | Instr. text | IL BART | Human expert (first) | Human expert (last) |
|---|---|---|---|---|
| Score | 79.7 | 83.0 | 82.1 | **84.4** |
| Success Rate | 52.6% | 57.6% | 57.9% | **61.0%** |

Table 4: Task performance with the Choice oracle. *first* and *last* refer to the first and last search queries found in human demonstrations, respectively.

**matching**, which is important in choosing options that contain noisy paraphrases of instruction spans. In the second example, the human explores several products first, and decides to return to the first explored product, demonstrating long-term **memory** that is lacking in the IL+RL model.

**Effect of RL fine-tuning after IL.** Table 2 also shows that RL fine-tuning adapts the IL model to become more 'greedy' and less 'exploratory', as the average trajectory length drops from 9.4 to 4.8, and the model explores fewer items and search queries. As a result, the attribute, type, and price scores all increase, but option score drops from 45.2 to 38.9. This points to the need for a better balance exploration with exploitation during RL, e.g. by using intrinsic bonuses.

**Results with at Choice oracle.** To disentangle the effects of learning to search from choosing the right actions, we construct a Choice oracle that has access to the hidden reward function as well as hidden attributes and options underlying each product and instruction.[†] Given a search query, the Choice oracle will perform an exhaustive search over every result item, try out all available options and finally choose the best item with options that maximize the reward — meaning each episode will take more than a hundred steps, as opposed to 4.5 and 11.3 steps on average for the IL+RL model and human experts (Table 2). We use 500 test instructions and consider four types of search queries: the instruction text (used by rule baseline), top IL BART generated query (used by all learning models), and the first and last queries from human experts in each test trajectory.[‡] Choice improves the success rate of rule heuristics from 9.6% to 52.6%, and the IL model from 29.1% to 57.6% (Table 4), confirming that choosing the right actions is indeed a major bottleneck for current

---

[†]A similar search oracle is also possible but harder to design since the search space is infinite. One possible oracle is to search for the underlying product name for each instruction, but that makes choice trivial as the underlying product is then almost always the first search result.

[‡]74.8% of the time there is only one query in the trajectory.

models with great room for improvement. However, it does not impact human performance much since they are likely good at making good choices.

### 5.4 Zero-shot Sim-to-real Transfer

Finally, we conduct a '*sim-to-real*' transfer experiment where our models trained on WebShop are tested on the real-world Amazon (`amazon.com`) and eBay (`ebay.com`) shopping websites without any fine-tuning. We sample 100 test instructions and deploy 3 WebShop models (rule, IL, IL+RL) to interact with Amazon and eBay, and manually score each episode based on Eq. (1). As shown in Table 5, model performances on the two website are similar to WebShop performances in Figure 4, except for the rule baseline, likely due to the better search engine of Amazon than WebShop.

| | Amazon | | | | eBay | | | | |
| | Score / SR | Att | Opt | Type | Price | Score / SR | Att | Opt | Type | Price |
| --- | --- | --- | --- | --- | --- | --- | --- | --- | --- | --- |
| Rule | 45.8 / 19% | 45.6 | 38.0 | 66.2 | 90.0 | 31.7 / 7% | 62.3 | 25.9 | 49.0 | 67.0 |
| IL | 61.5 / 27% | 60.7 | **53.7** | 85.6 | 96.0 | 58.2 / **21%** | 60.2 | **52.3** | 85.1 | 96.9 |
| IL+RL | **65.9 / 25%** | 71.6 | 47.0 | **87.8** | **100.0** | **62.3 / 21%** | 69.1 | 39.5 | **91.7** | **97.0** |
| Human | 88.2 / 65% | 86.2 | 76.3 | 99.0 | 100.0 | 79.7 / 40% | 80.3 | 70.1 | 99.5 | 100.0 |

Table 5: Zero-shot sim-to-real transfer to Amazon and eBay over 100 test instructions. The Score / SR (Success Rate) column indicates the overall performance. The remaining breakdown are in Score.

On `amazon.com`, IL+RL achieves a Score of 65.9 and SR of 25%, outperforming the Rule baseline's Score of 45.8 and SR of 19% by large margin. Similarly, on `ebay.com`, IL+RL achieves a Score of 62.3 and SR of 21%, widely outperforming the Rule baseline's Score of 31.7 and SR of 7%. These results confirm positive sim-to-real values of trained agents for real-world web tasks despite domain shifts in data (products) and dynamics (search engine). We also obtain a human average score of 88.0 / 79.7 and success rate of 65% / 40% by asking turkers (§3.2) to find the instructed product on the Amazon and eBay websites respectively. While humans perform much better than agents, their web interactions are much slower — taking on average 815 seconds per episode as opposed to $< 8$ seconds per episode for our IL and IL+RL models on Amazon. This sim-to-real transfer only requires two minor coding additions, suggesting that environments like WebShop are suitable for developing *practical* grounded agents to reduce human effort on real-world web tasks. We provide additional performance and in-depth analysis in Appendix §D.

## 6 Discussion

We have developed WebShop, a new web-based benchmark for sequential decision making and language grounding, modeled on interaction with an e-commerce website. We performed an empirical evaluation of autonomous agents trained using imitation and reinforcement learning, and demonstrated promising results on sim-to-real transfer to real-world shopping websites. Our qualitative and quantitative analysis of model and human trajectories (§5.3) identified several research challenges in WebShop and provided insights for future model development by incorporating multidisciplinary techniques. For example, pre-training with multi-modal data [27, 52], web hypertext [2], or web instruction-action mapping [38] could help agents better understand and leverage rich semantics of webpage content, actions, and instructions. Ideas from query (re)formulation [22, 59, 37, 51] may help agents expand the range of search exploration, and improved action exploration [40, 12, 49] and memory [53, 13, 23] mechanisms could help agents make better decisions over the long horizon and large action space. The modular design of WebShop also allows for new web tasks and domains to be easily incorporated, which we hope will help shape future research into grounded language agents with stronger capabilities for real-world web interaction.

## Acknowledgements

We thank Alexander Wettig, Ameet Deshpande, Austin Wang, Jens Tuyls, Jimmy Yang, Mengzhou Xia, Tianyu Gao, and Vishvak Murahari from the Princeton NLP Group for proofreading and providing comments. This material is based upon work supported by the National Science Foundation under Grant No. 2107048. Any opinions, findings, and conclusions or recommendations expressed in this material are those of the author(s) and do not necessarily reflect the views of the National Science Foundation.

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
