# A  Environment Details

## A.1  Product Scraping

We use ScraperAPI [35] to extract publicly available product information from `amazon.com`. We use five categories (beauty, food, fashion, furniture, electronics) and 313 associated sub-category names appeared in `amazon.com` (e.g. "Women's Loafers & Slip-Ons" in fashion, "Pendants and Chandeliers" in furniture) to scrape $1,181,436$ products. We filter products with duplicate titles or product IDs, but do not perform extra filtering in order to avoid selection bias. Specifically, as `amazon.com` has its own content screening process, we did not find any personally identifiable information or offensive content during random sampling checks.

| Products | Unique Attributes | Avg Attributes | Unique Options | Avg Options |
|----------|-------------------|----------------|----------------|-------------|
| 1,181,436 | 670 | 3.1 | 842,849 | 0.67 |

Table 6: Product item statistics.

## A.2  Product Attribute Mining

We use `TfidfVectorizer` from `scikit-learn` to extract probable bi-grams as attributes from product title and descriptions for further annotation. We manually inspect these attributes to keep only the *specific* and *human-readable* ones and filter out the rest. An attribute should be suitable in at least one of the following use: 1) `IsGoodFor`, 2) `HasA` (contains), 3) `WhichIs`, and 4) `IsA`. For example, attributes such as "oz ml" and "men women" will be filtered out since it's unparsable. On the other hand, "hair color" will also be filtered since it is not specific enough to fit in the above 4 categories. Attributes such as "dry skin" can fit the `IsGoodFor` in the context of a make-up product being good for dry skin.

## A.3  Search Engine

Each time the agent performs a search, the top 50 items are retrieved and displayed across five search result pages, where each page contains 10 items and the agent can use actions choose[Prev/Next page] to navigate across result pages. Figure 2 shows that when searching directly with the instruction text, the corresponding item appears in the first search page (rank 1-10) nearly 1/3 of the time, but it cannot be found in any search pages (rank 50+) more than half of the time. This indicates that while the search engine can decently retrieve items based on lexical matching, directly searching the instruction is not enough for solving the task, and good query (re)formulation based on the instruction is important.

## A.4  Instruction Collection

We collect human written instructions by providing the workers a product including the title, product category, and its set of attributes and options (Figure 5, 6). We conduct qualification task by having each participating workers to work on $2-5$ examples. We inspect and assign qualification to 213 workers to perform the instruction writing task. We pay for each example $0.15$ dollars. We do not anticipate any potential participant risk.

## A.5  Reward Calculation

The type reward $r_{\text{type}}$ consists of 3 elements: 1) course-grain product category match ($c = 1$ if matched), 2) fine-grain category match ($f = 1$ if matched), and 3) product title match. Course-grain product category refers to the 5 categories described in §3.2. Fine-grain category is the chain of categories that the product is under on the Amazon website. For example, and eye mask sheet would be under the *Beauty & Personal Care > Skin Care > Eyes > Wrinkle Pads & Patches* fine-grain category. The product title refers to $\bar{y}$ described in §3.

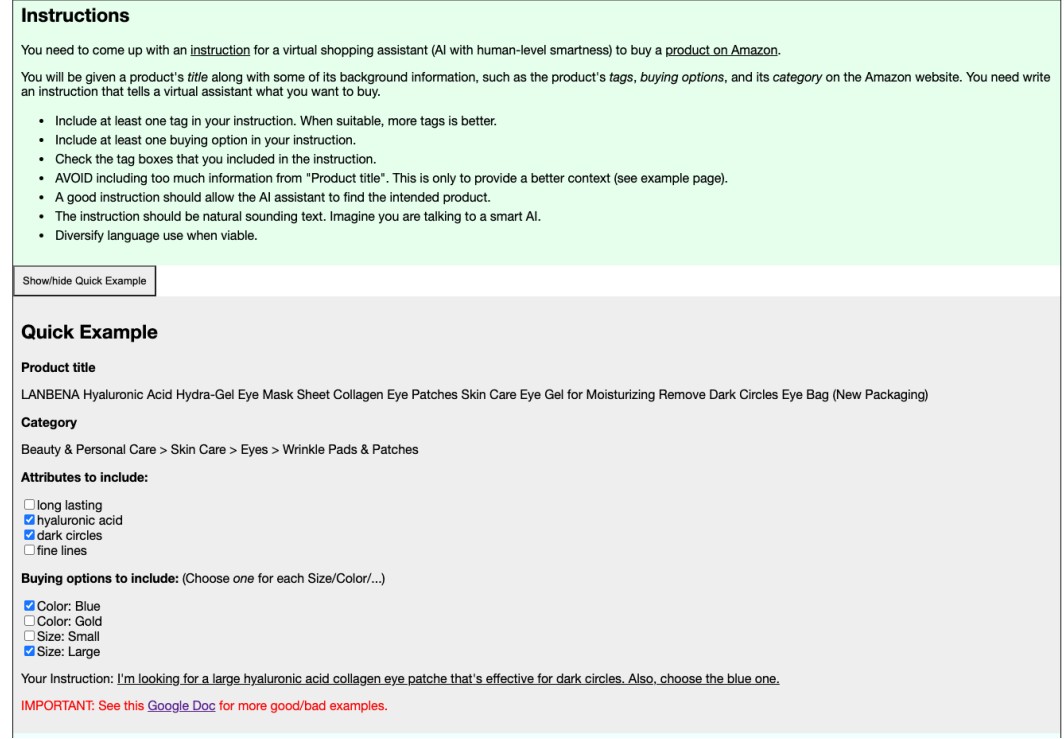

Figure 5: The Amazon Mechanical Turk interface for the instruction writing task. The green box shows the general instruction for the task and the grey box shows an concrete example.

**Your Task**

**Product title**

Applicator, Sturdy Stable Compact Portable Ergonomic Hair Removal Cream Spatula for Cosmetics Shop for Home for Beauty Salon

**Category**

**Attributes to include:** (Choose at least one. Choose more when applicable.)

- ☐ easy apply
- ☐ eco friendly
- ☐ non toxic
- ☐ high quality
- ☐ hair removal
- ☐ beauty salon

**Buying options to include:** (Choose one for each Size/Color/...)

UPDATE 04/11/2022: Feel free to paraphrase the tags and options to make the instruction sound more natural.

UPDATE 04/13/2022: Please don't copy the entire title exactly. Just include necessary info from the title to know what kind of product it is.

Input your instruction here...

Any comment or feedback? (optional)

**Submit**

Figure 6: The Amazon Mechanical Turk interface for the instruction writing task. The blue box shows the actual annotation interface. The worker is required to check the boxes and write the instructions in the text field before submission.

$$
r_{\text{type}} = \begin{cases} 0, & \text{if } \texttt{TextMatch}(\bar{y}, \bar{y}^*) = 0 \\ 0.1, & \text{if } \texttt{TextMatch}(\bar{y}, \bar{y}^*) < 0.1 \\ 0.5, & \text{if } \texttt{TextMatch}(\bar{y}, \bar{y}^*) > 0.2 \text{ and } c = 1 \text{ and } f = 1, \\ 1, & \text{otherwise} \end{cases} \tag{6}
$$

| Instruction 1: I would like a stained glass wall lamp with a bronze finish, and price lower than 190 dollars. | Instruction 2 I would like a lead free bracelet birthday cake jar candle, and price lower than 50.00 dollars. |
|---|---|
| Human Actions ($r = 0.33$, length = 4) search[stained glass wall lamp] click[item-QCLU Tiffany Style Lamp Sunflower...] click[wall lamp 3 - 12 inch] click[buy] | Human Actions ($r = 0.03$, len = 4) search[lead free bracelet birthday cake jar candle] click[item-Happy Birthday Candle...] click[8 ounce round tin] click[buy] |

Table 7: Two examples of failed human trajectories. A common pattern is impatience: after one search (even with correct attributes like the right example) the less performant worker commits to the first selected item. Often, the item does not contain the desired options even though the item's title text seem relevant. An expert worker will recognize the need to select the correct options and go back to refine the searches, while less performant workers simply commit to the current selected item.

Here, TextMatch($\bar{y}, \bar{y}^*$) is a simple string match between the selected product title text and the goal product title text. We use only the words tagged with PNOUN, NOUN, and PROPN tags parsed by the SpaCy parser in the title text.

## A.6 Human Trajectory Collection

We use the HTML environment in Figure 1 to collect human trajectories. We select a pool of 13 workers using qualification tasks where each workers complete 5 examples. The workers that achieve an average reward more than $0.75$ are qualified. The task instruction is shown at the end of Appendix. We observe a pronounced performance gap between the very high performing workers and average workers. We use the top $50\%$ of these qualified workers as experts (7 workers in total). We pay for each completed trajectory $0.7$ dollars. In human evaluation, 8 out of the 13 workers participated and 5 among them are in the aforementioned expert pool. The 8 participants achieve an overall score of $75.5$ and a success rate of $50.0\%$ We observe non-negligible variance even within the experts—the best performer achieves a score of $87.4$ and success rate of $69.5\%$, while the lowest performing worker achieves a score of $45.8$ and success rate of $10\%$. The best performing worker also shows better consistency—drawing at a standard deviation of $2.3$ in score, contrasting the lowest performing counterpart at $3.1$. We provide examples of common human failure cases such as not matching the option/attribute due to impatience (Table 7), cautioning some caveats of the task with human workers.

## A.7 Reward Verification

We randomly select 100 samples each from the pools of trajectories generated by average and expert MTurk workers. Each trajectory is then manually re-scored against a human criteria; the purpose of this is to determine how representative the reward function is of a human's judgment towards whether the chosen product satisfies the given instructions. The human score calculation procedure exactly follows the formula laid out in Section A.5 – the attribute, option, price, and type scores are individually determined, then aggregated to calculate the overall score – except for one main modification. Instead of the exact matching approach, points are awarded if (1) the picked product's attributes, options, or type are lexically similar or synonymous with the goal's product information and (2) the desired value is not found verbatim anywhere in the picked product's descriptions. For instance, if the value *lightweight* is specified as a desired attribute for an instruction, but the value *easy carry* is found instead in the picked product's description, then the attribute score for the picked product is increased to reflect that the *lightweight* value was found. On the other hand, if *cyan* is desired as an option for a goal product, but the user picks *blue* even though *cyan* is available as a choice, then no points are awarded. To ensure the score is calculated without bias, the original rewards for each trajectory were not compared with the human evaluation scores until the human evaluation scoring was completed.

For the average trajectories, the automatic task score was $74.9$ and our manual score was $76.3$ with a Pearson correlation of $0.856$. For expert trajectories, the respective scores were $81.5$ and $89.9$ with a Pearson correlation of $0.773$. Therefore, the automatic reward seems to provide a reasonably close lower bound to the actual task performance. We find that for average workers, $87.0\%$ of automatic

scores are within a $10\%$ of the manual score, with the main source of error being synonyms or lexically similar words that don't get matched correctly in the automatic reward function.

| MTurk Type | Reward Function | Price | Type | Attribute | Result | Overall |
|---|---|---|---|---|---|---|
| Average | WebShop | 95.0 | 92.9 | 71.7 | 50.5 | 74.9 |
| | Human | 95.0 | 93.8 | 75.3 | 57.0 | 76.3 |
| Expert | WebShop | 100.0 | 100.0 | 78.1 | 56.1 | 81.5 |
| | Human | 100.0 | 100.0 | 88.2 | 66.8 | 89.9 |

Table 8: Reward Verification Statistics

Table 8 reflects our observation that our reward function is similar to a human's score, with a consistent tendency to over-penalize the picked product. For every trajectory's product, the human score across all categories (e.g. attributes, options) is always greater than or equal to the original score. This under-scoring is a result of our reward function's exact matching criterion. In future work, we hope to improve our matching functionality such that, within the context of a single product with respect to the goal instructions, it can identify synonyms and decide whether to award additional points.

# B  Model Details

## B.1  Cross Attention Layer

Our cross attention layer follows Seo et al. [42]. Denote the $i$-th contextualized token embedding from the observation and action to be $\mathbf{o}_i$ and $\mathbf{a}_i$ respectively. The attention between $\mathbf{o}_i$ and $\mathbf{a}_j$ is defined as

$$\alpha_{ij} = \mathbf{w}_1 \cdot \mathbf{o}_i + \mathbf{w}_2 \cdot \mathbf{a}_j + \mathbf{w}_3 \cdot (\mathbf{o}_i \otimes \mathbf{a}_j) \tag{7}$$

where $\otimes$ denotes element-wise product and $\mathbf{w}_1, \mathbf{w}_2, \mathbf{w}_3$ are learnable vectors. The observation-contextualized vector for $j$-th action token is then

$$\mathbf{ca}_j = \mathbf{w}_5 \cdot \text{leakyRELU}(\mathbf{w}_4 \cdot [\mathbf{a}_j, \mathbf{c}_j, \mathbf{a}_j \otimes \mathbf{c}_j, \mathbf{q} \otimes \mathbf{c}_j]) \tag{8}$$

$$\mathbf{c}_j = \frac{\sum_i \exp(\alpha_{ij}) \cdot \mathbf{o}_i}{\sum_i \exp(\alpha_{ij})}, \quad \mathbf{q} = \frac{\sum_{j'} \exp(\max_i \alpha_{ij'})\mathbf{a}_{j'}}{\sum_{j'} \exp(\max_i \alpha_{ij'})} \tag{9}$$

We then average pool all $\mathbf{ca}_j$ to derive the action score $S(o, a)$:

$$S(o, a) = \mathbf{w}_6 \cdot \frac{1}{n_a} \sum_{j \leq n_a} \mathbf{ca}_j \in \mathbb{R} \tag{10}$$

where $n_a$ is the number of tokens for action $a$.

## B.2  RNN Baseline

Our RNN baseline is inspired by Guo et al. [14], where we use the same attention layer as described above, but replace the Transformer text encoder with one-layer bi-directional Gated Recurrent Units (GRU) [9] of hidden dimension 512. Another difference is that we also add an cross attention between the instruction and action input word embeddings, as we hypothesize it might help option text matching.

# C  WebShop Experiment Details

## C.1  IL Training Details

The training code for our IL models is adapted from Huggingface glue training example, whose repository is licensed under Apache License 2.0. We use a training batch size of 1 with 32 gradient accumulation steps, a learning rate of $2 \times 10^{-5}$, and 10 training epochs. The training takes around 2 hours on one RTX 2080 GPU with a GPU memory of around 10GB.

|  | Score | SR |
|---|---|---|
| IL | 60.56 (1.94) | 29.00 (2.42) |
| IL (top-1 search) | 61.96 (0.47) | 30.80 (0.72) |
| IL (top-1 choice) | 45.10 (3.50) | 24.93 (3.14) |

Table 9: Sampling vs. top-1.

|  | Score | SR |
|---|---|---|
| IL | 60.6 (1.94) | 29.0 (2.42) |
| IL (w/o image) | 60.3 (0.47) | 28.4 (0.87) |

Table 10: Image ablations.

## C.2 RL Training Details

We train the RL models using 4 parallel environments for $100,000$ training steps. The backprogation through time (BPTT) is taken at every 8 steps. We use an Adam optimizer with a learning rate of $10^{-5}$ (for Transformer models) or $5 \times 10^{-4}$ (for RNN models).

For RL models with the Transformer (BERT) architecture, it takes around 27 hours on one RTX 3090 GPU with a GPU memory of around 20GB. For RL models with the GRU architecture, it takes around 20 hours on one RTX 2080 GPU with a GPU memory of around 10GB.

To disentangle the effects of learning to search from choosing the right actions, we construct a `Choice` oracle that has access to the hidden reward function as well as hidden attributes and options underlying each product and instruction.[§] Given a search query, the

## C.3 Sampling vs. Top-1

We show comparisons between using beam search vs. top-1 for both the search model and the choice model in Table 9. During testing, the search model uses beam search to generate top-5 search queries. We randomly and uniformly sample from the top-5 queries to increase search diversity in case of multiple searches. We conduct experiments to instead always use the top-1 search, which shows slight performance improvement (see table below), and we will include the result in the paper. The choice model has a fixed set of action candidates at each step (e.g. all available buttons), and we sample from the choice policy what action to take, as always taking the top action will lead to significantly detorior performances.

## C.4 Image Ablation

We train 3 trials with different random seeds for both the IL model and the ablated IL model without images, with performances over 500 test cases (10). Removing image only slightly hurts the overall performance, but significantly reduces the variance. This is reasonable as our current instruction and reward setups only use textual information, and we believe future efforts to incorporate visual information into the task setup will better challenge models' visual understanding, and make pre-trained vision-language models such as CLIP more useful.

# D  Sim-to-real Details

## D.1  Sim-to-real Transfer Details

To test how well our IL agent trained in WebShop performs on `amazon.com` (`ebay.com` similarly), we wrote a series of scripts that generally achieve two steps - translate a real Amazon URL into our IL model's input (text observation, set of valid actions) and map the model's output back to a real Amazon URL. The following procedure is repeated until the IL model generates a "buy now" action:

---

[§]A similar search oracle is also possible but harder to design since the search space is infinite. One possible oracle is to search for the underlying product name for each instruction, but that makes choice trivial as the underlying product is then almost always the first search result.

- Amazon URL → Amazon HTML → Amazon Page Information: Using ScraperAPI [35], we first get the HTML source code for a given Amazon page, then extract information relevant to rendering the equivalent page in the WebShop environment (e.g. title, price, options).

- Amazon Page Information → WebShop HTML → Text Observation: Given the scraped information, we generate the corresponding WebShop page in HTML mode, then transform it into a `simple` mode text observation.

- Amazon Page Information → Valid Action Set: From the scraped information, we determine what valid actions the model can take (i.e. `search[Red shoes]`, `choose[Size 9]`). This logic is captured as a mapping of page type to permissible actions.

- Text Observation, Valid Action Set → IL Model → Amazon URL: Given the text observation and allowed of valid actions, the IL model produces an action. This action is then used to construct a corresponding Amazon URL via a set of mapping rules, and the loop is repeated. This continues until the model generates a "buy now" action, terminating the loop.

## D.2  Sim-to-real Transfer Results

The resulting numbers in Table 5 closely cohere to the reported numbers of WebShop found in Figure 4, suggesting that the WebShop has promise for developing grounded agents that can operate on real web environments. Between the two websites, transfer to Amazon is better than eBay as we note that (i) eBay has a larger product gap from WebShop, e.g. some item categories like food are disallowed in eBay. (ii) the eBay search engine seems weaker, and would sometimes display no results for lengthy instructions. The following Table 11 is an example of a trajectory generated by the IL agent searching on the real Amazon website.

---

**Instruction**: I want to find white blackout shades that are 66 inches in width and 66 inches in height. They need to be easy to install..

search[white blackout shades 66 inches in width and 66 inches height, easy to install] click[item - Easy Up & Down 100% Blackout Pleated Window Shades Temporary Window Blinds 36in x 64in (Fits Window Width 18"-36") 2pcs-Pack Operating with Pull Cord Easy Trimming & Installing] click[features] click[back to search] search[white blackout shades that are 66 inches in width and 66 inches height] click[item - Redi Shade Inc 1617201 Original Blackout Pleated Paper Shade Black 36" x 72" 6-Pack] click[< prev] click[Shade + Strips, White] click[buy]

---

Table 11: An example trajectory (showing only actions) from the IL agent on the real Amazon website. We omit instructions and some human actions for instruction and trim item names for readability. Red denotes options and blue denotes attributes.

It is evident that the exploratory behavior and patterns learned and exhibited by the agent within the WebShop environment is not lost in this transfer. These results point to the opportunity for sim-to-real trained agents to transfer to other real-world web tasks despite the domain shift in both data (products) and dynamics (search engine) With that said, the gap between human and model performance also encourage us to look into expanding on the current limitations in our work regarding both the model and the WebShop environment.

## E  Potential Societal Impacts and Limitations

WebShop is designed to minimize human efforts in data collection and processing, but there are still potential concerns regarding diversity, fairness, and representation. Developing RL agents that interact with the web also bear safety concerns, especially when transferring from simulation to real-world websites. We also discuss other limitations regarding the semantics of current task (instruction/reward).

**Diversity and representation in data collection.** We chose five common categories from amazon.com and scrape all products using all subcategories to minimize bias. However, our data is still biased toward the website country (USA) and website language (English), and may only represent a subset of all possible products that users potentially want to buy. Having this limitation in mind, the design of WebShop allows the product data to be easily updated for different representations of real-world usage.

**Bias in data processing.** Currently our attribute labeling is manually done and may be biased by the labeller's own experience (e.g. more knowledge toward product attributes like sports rather than makeup). An more automatic alternative would be to employ trained NLP models (e.g. relation extraction) to extract product attributes, which might be less biased than one labeller. Our reward design is general and could be updated to weight more toward attributes, options, price, etc.

**Safety for developing web agents.** Unlike recent work [33] that directly employs agents on the World Wide Web (WWW), WebShop aims to provide a realistic simulation environment to train agents in a controllable and safe manner. In our preliminary sim-to-real experiments, the agent could only update the current webpage's url in two fixed and safe ways (i.e. search for results, open an item), and any form sending action (e.g. click options or buy) is held within the sim-to-real interface for later reward calculation. As a result, only navigation is done on the real-world website. For future deployment to real-world websites with more advanced functions, we believe a good specification of possible model behaviors is key to avoid harmful actions.

**Limitations in the current task.** Our current instructions are still limited by the attributes and options used. While attributes are simple and sometimes too generic (e.g. "easy to use"), the options might get too specific (e.g. "d17(dedicated right, back)"). Therefore, an agent might sometimes use a special option as cues to find the product, while ignoring other parts of the instruction. To better leverage images and texts (including reviews written by human users, which are not used in current work) of products for more semantic and challenging instructions is an important future direction from WebShop.

## Instruction for Human Trajectory Collection

The following pages display the human trajectory collection document mentioned in §A.6.

# The WebShop Task

Thank you for taking part in this project! In this task, you need to buy a designated product given an instruction on our Amazon Shopping Game site. You will get a score in the end indicating how close you are. Please try to score as high as you can.

If you find in some cases the scoring seems weird/unfair, please reach out. We will look into the cases.

Please read the following instructions carefully before you start.

## Instructions

**1**) Go to the home page. The instruction will immediately show up on the landing page.

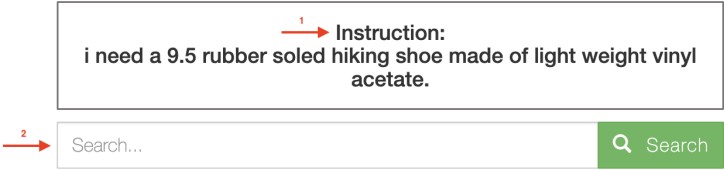

**2**) Given this instruction, please write a search query that would produce search results matching the description.

Please *do not* copy-paste the entire instruction. We encourage you come up with more targeted queries, see the result, and search again if needed.
Example:

- `Instruction`:
  I need a 9.5 rubber soled hiking shoe made of lightweight vinyl acetate.

- `Bad query`: (copy pasting)
  9.5 rubber soled hiking shoe made of lightweight vinyl acetate

```
- Ideal query: (1st attempt)
  rubber soled hiking shoe vinyl acetate (say the results are not great)
- Ideal query: (2nd attempt)
  hiking shoe lightweight vinyl acetate  (the results are better)
- Ideal query: (3nd attempt)
  lightweight climbing shoe vinyl        (gives promising results)
```

Essentially, you need to hack the search engine a little bit.

Note that our search engine is limited. Tricks that work on Google Search such as adding quotation marks around the query won't work.

Click **Search** after filling out the search bar like below.

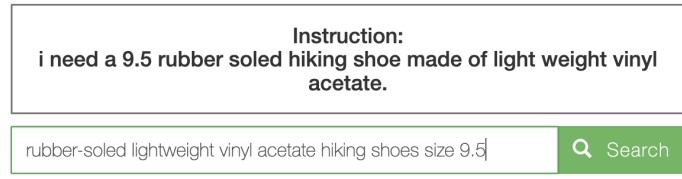

**3**) Upon clicking **Search**, you will be sent to a page of results. The below screenshot is an example of the results displayed from the example query in Step 2. Each page shows up to 10 results. Click the **Next** button to see more results.

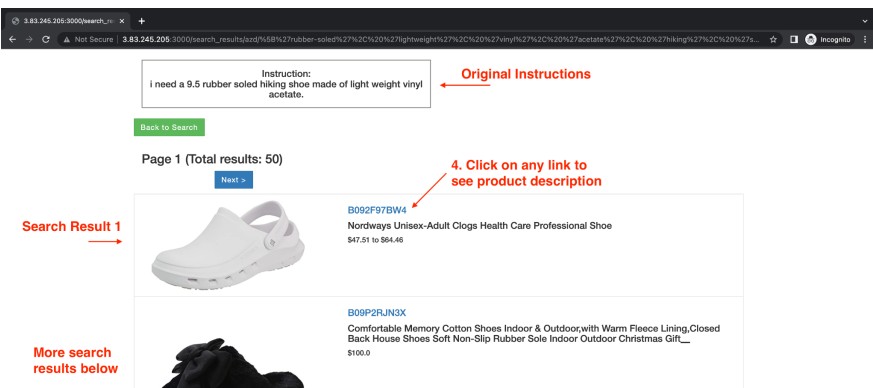

**4**) Click on any of the blue product title text (i.e. *"B092F97B24"* in above screenshot) to see a product detail page, like the below.

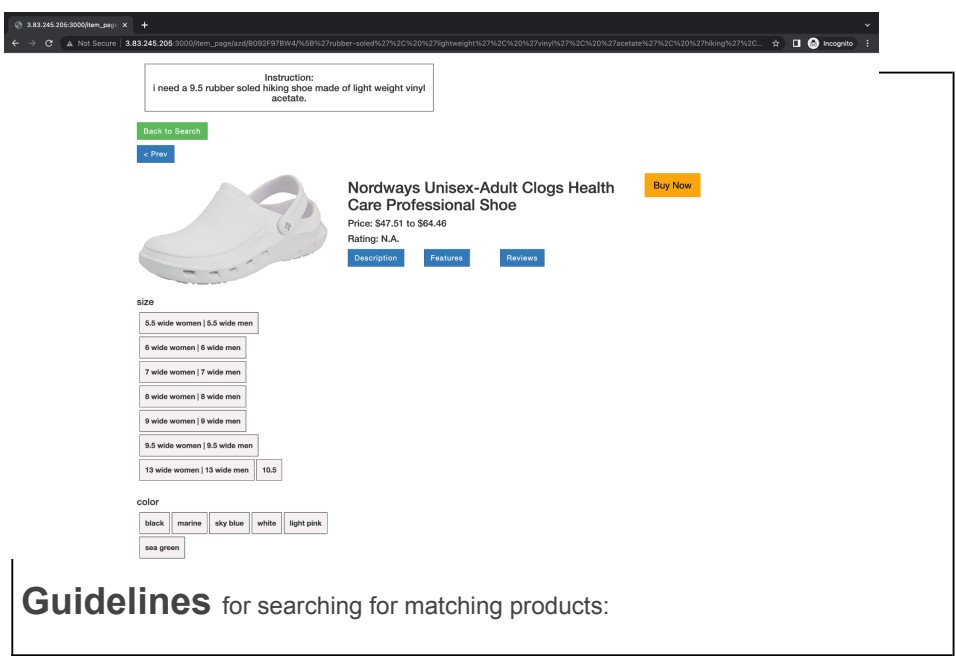

**Guidelines** for searching for matching products:

- Some pages have **Options** (i.e. *Size*, *Color* in above screenshot). If the instructions contain such information, please select the corresponding options (even if the title / features / desc. / reviews may already contain such info). In most cases, if you find the options verbatim as in the instruction, you've likely found the right product.
- Do **not** use the product image to determine whether the instruction's information matches the product.

    An example:
    - Given instruction: "Find me a pair of ankle socks that are blue and size 11"

Between this product…                          And this product…

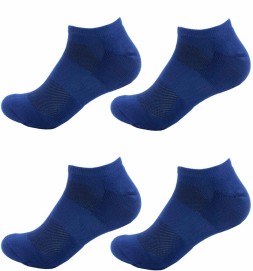                          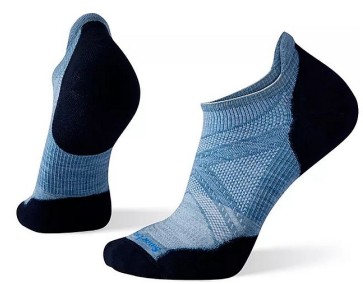

**TITLE**: Ankle socks for casual wear, sports, and leisure. Pack of 4, 8, or 12

**DESC**: 100% made in the USA. These socks are good for any occasion.
**FEATURES**: Made with cotton, breathable fabric. Machine washable okay.

**OPTIONS**:
- Sizes: 8, 9, 10, 11, 12
- Color: red, green, black, white, blue

**TITLE**: Kirkland athletic socks with rubber soles and heels. Easy slip on

**DESC**: Costco wholesale socks, limited stock.
**FEATURES**: Polyester and Rayon fabric. Guaranteed long lasting or your money back.

**OPTIONS**: None

The left hand is a **better match** because the product's title, features, description, and options reflect the instruction's information.

While the right hand product appears to be a pair of blue ankle socks, because this information is not reflected in the text, we do **not** consider this a match.

> Therefore, feel free to use the product image as a reference when looking for matches, but keep in mind that the experiment we're running accounts for a text's

**5**) Decide whether the product is a match

> A **match** should
> - Contain all of the instruction's information in the product detail page's text (i.e. title, description, feature, options)
> - Have options (if they exist), which correspond to the product info, be selected.
>
> A match does **not** account for
> - The product image

- *You think it is a match!* → Click the Buy Now button on the product detail page
- *You think it is not a match OR another product might be a better match…*
  - Click on the Back button to go to the original list of search results (page **3**). From here, repeat steps **3-4** until you find a product that matches best.
  - Click on the **Back to Search** button. This will take you back to the search bar page (page **2**). If you feel none of the results are good matches, try another search query.

**6**) Once you clicked Buy Now, you will see your score (won't be used to decide the pay), and a code you need to paste in the MTurk interface. And you're done!

# Tips
Patterns that often result in HIGH scores:
- Refine search queries until promising products show up
- Explore different product pages (go to next page if needed) to see if options and different aspects are covered
- Make sure all aspects in the instructions are covered by either the title, description page, or the feature page.
- Make sure all options are found almost verbatim in the product page

Patterns that often result in LOW scores:

- Low effort copy-paste the entire instructions as the search query
- Always click the first item without checking if the aspects in the instructions are covered
- Click items that obviously don't have any option matches