# OpenReview forum: "WebShop: Towards Scalable Real-World Web Interaction with Grounded Language Agents"
_NeurIPS.cc/2022/Conference — NeurIPS 2022 Accept_

### Official Review · Reviewer_ytjW · 2022-07-09

**Rating:** 7
**Confidence:** 3
**Soundness:** 3 good
**Presentation:** 4 excellent
**Contribution:** 3 good

**Summary:**

- The authors collect and release a dataset, WebShop, designed to train and evaluate agents in a web-based interaction task. The dataset contains natural language requests for products that match various attributes, options, and price thresholds, and subsequent human trajectories when navigating an interactive environment in which one can search, click through product pages, explore item details, look at item images, and ultimately purchase a product.
- The authors propose two baselines: 1) a rule-based one, in which the request is fed to the search engine and the first product is picked; 2) a suite of models that uses BART to generate search queries, and ResNet and Transformer models to processes item images and text in order to predict actions and purchases. The latter method is trained using both imitation learning (w.r.t. Human trajectories) and reinforcement learning. The authors find that the proposed method outperforms the rule-based baseline, but underperforms humans by quite a large margin. Finally, the authors analyze where the gaps between human and model performance lie, and suggest various modeling improvements.

***EDIT AFTER REBUTTAL PERIOD***
Revising my score to 7; see below comments

**Questions:**

- Did you analyze why human performance is so low? What is preventing humans from finding the correct products? Or is this simply the nature of human purchase behavior?
- What is LP? It is mentioned in section 5.1 but is not properly introduced (e.g., “IL (w/o LP Choice)”)
- Was a model with full imitation learning trained? If not, why not?


**Limitations:**

The authors adequately address the limitations in the conclusion.

**Strengths And Weaknesses:**

### Originality
- **Strengths**: The authors propose a novel benchmark that can be helpful to the community - indeed the closest related work (as noted) is WebGPT, though this task differs in a few ways (e.g., including images). It is clear how the work differs from other contributions.

### Quality
- **Strengths**: Claims are well supported by empirical results; the authors clearly enumerate the axes on which scores differ across methods and humans, and from this can conclude various ways in which to improve the methods. The proposed model, while simply an amalgamation of prior-used techniques, is a nice starting point for the benchmark.
- **Weaknesses**: The authors, in my opinion, missed a few key baselines, and perhaps overstate the performance of their proposed method. While the contribution of the paper is not the method itself, but rather the dataset, I don’t think comparing it to the rule-based baseline is a proper comparison (as it essentially compares to just a BM25 retrieval performance vs. three neural network models). I also don’t fully understand why the authors omitted a fully-trained IL baseline, as they only used 1/10th of the dataset to train the IL model. Additionally, some other design choices are not quite justified empirically (e.g., using sampling instead of true beam-search)

### Clarity
- **Strengths**: Claims are well supported by empirical results; the authors clearly enumerate the axes on which scores differ across methods and humans, and from this can conclude various ways in which to improve the methods. The proposed model, while simply an amalgamation of prior-used techniques, is a nice starting point for the benchmark.
- **Weaknesses**: The authors, in my opinion, missed a few key baselines, and perhaps overstate the performance of their proposed method. While the contribution of the paper is not the method itself, but rather the dataset, I don’t think comparing it to the rule-based baseline is a proper comparison (as it essentially compares to just a BM25 retrieval performance vs. an entire neural network). I also don’t fully understand why the authors omitted a fully-trained IL baseline, as they only used 1/10th of the dataset to train the IL model. Additionally, some other design choices are not quite justified empirically (e.g., using sampling instead of true beam-search)


### Significance
- **Strengths**: The public release of such a dataset is indeed important for those researching interactive web-based agents; if the presence of related work is any indication, this is an important direction for many. The authors also provide several “low-hanging” improvements to the model based on analyses of their baseline.
- **Weaknesses**: Human performance on the task is, in my opinion, quite low, and brings into question the validity of the dataset itself - how useful can it be if humans cannot even reach 60% task performance? This, to me, is the greatest weakness of the work.

---

> ### Author Response · Authors · 2022-08-01
> **Thank you - Individual Response to ytjW**
>
> 1. **Overstate the performance of their proposed method (when the baselines it compares to are weak)**
>
> The rule heuristic actually achieves a non-trivial attribute reward mainly relying on BM25 doing the heavy lifting. While IL and IL+RL models indeed significantly outperform it, we also find IL without language pre-training or RL without IL warmstarting will lead to similar performances to the rule baseline (Figure 4), showing non-triviality and diagnostic value of the rule baseline for modeling. Like the reviewer mentions, our goal is not to achieve the start-of-the-art on our own benchmark. Instead, our experimental results using a combination of current methodologies (e.g. rule, IL, RL) (Section 5.1) and various ablations (Section 5.2) and analysis (Section 5.3) aim to provide comprehensive modeling insights for future development of more advanced web agents.
>
> 2. **Fully-trained IL baseline**
>
> To clarify, the IL training uses the entirety of the 1,012 collected training trajectories. Due to resource limitations we could not crowdsource human trajectory for all 10, 587 training instructions, but the RL and IL+RL models do use all 10,587 training instructions thanks to the automatic reward function.
>
> 3. **Some design choices are not quite justified empirically (e.g., using sampling instead of true beam-search)**
>
> During testing, the search IL model uses beam search to generate top-5 search queries. We randomly and uniformly sample from the top-5 queries to increase search diversity in case of multiple searches. We also conducted experiments to always use the top-1 search, which shows slight performance improvement (see table below), and we will include the result in the paper.
>
> The choice IL model has a fixed set of action candidates at each step (e.g. all available buttons), and we sample from the choice policy what action to take, as always taking the top action will lead to significantly detorior performances (see table below).
>
> | | score | success rate (\%) |
> | -------- | -------- | -------- |
> | IL  |   60.56 (1.94)  |  29 (2.42)   |
> | IL (top-1 search)  |  61.96 (0.47)     | 30.8 (0.72)  |
> | IL (top-1 choice)  |  45.10 (3.50)     | 24.93 (3.14)  |
>
>
> 4. **How useful can it be if humans cannot even reach 60% task performance? This, to me, is the greatest weakness of the work. Did you analyze why human performance is so low? What is preventing humans from finding the correct products? Or is this simply the nature of human purchase behavior?**
>
> We thank reviewer ytjW for raising this important question. We have revised the draft to better address this. To summarize, there are several reasons for (relatively) lower human performance:
>
> 1. *Our Success Rate metric is strict by design*: Success Rate is a particularly harsh metric as it requires *all* attributes and options to be matched. Besides Success Rate, we also report Score (reward), which is a less harsh metric and human experts can achieve a score of 82.1 out of 100. This is true for other standard datasets as well in NLP –  for example, in HotPotQA (Yang et al., 2018; HotpotQA: A Dataset for Diverse, Explainable Multi-hop Question Answering), human annotators only achieve an answer-fact joint Exact Match (EM) of 52.30, but a joint F1 of 82.55.
>
> 2. *The task is inherently challenging for humans* (see updated draft section A.6 table 6 for analysis): WebShop is a cognitively loaded task for humans to perform when instructions are compositional or contain rare options. To find all matching options and attributes, humans need to go back and forth to inspect multiple items and refine searches as well as adapting to a search engine less powerful than Google or Amazon Search. The impatience sometimes impedes humans from finding the exact product and might settle on a decent but suboptimal one. We believe that this actually presents an opportunity for models to automate the tiring shopping process and alleviate human efforts.
>
> Despite the above, we reiterate that the task presents research challenges and opportunities for models, since a large gap still exists between human and model performances. This gap provides sufficient research challenges for the community and insights for building grounded agents in a much more realistic environment than its preceding work. We have also updated the paper to provide a detailed analysis about the achievability of the task (updated draft section A.6) and the faithfulness of our automatic reward with regard to human assigned rewards (updated draft section A.7).
>
> 5. **What is LP? It is mentioned in section 5.1 but is not properly introduced (e.g., “IL (w/o LP Choice)”)**
>
> LP stands for language pre-training, and the concrete setup is in Section 5.2 (IL Ablations). We updated the text to specify this more clearly.

---

> ### Author Response · Authors · 2022-08-08
> **Thank you again**
>
> Dear reviewer,
> Thank you again for your review -- please let us know if you have any remaining questions or concerns, so that we can address them before the deadline tomorrow. Alternatively, if you feel that your original concerns are addressed, we would appreciate your updating your evaluation to reflect that.

---

> ### Comment · Reviewer_ytjW · 2022-08-08
> **Response to Authors**
>
> Thank you for clarifying the issues surrounding human performance and additionally the question of training data for the IL baseline. I do believe my original concerns were addressed. Given these, along with the additional clarifications around sampling vs. top-1 generation, I am comfortable raising my score from 6 to 7

---

### Official Review · Reviewer_smJp · 2022-07-11

**Rating:** 6
**Confidence:** 4
**Soundness:** 2 fair
**Presentation:** 3 good
**Contribution:** 3 good

**Summary:**

This paper focuses on the problem of language-grounded web interaction. The authors developed a simulated e-commerce website environment, i.e. WebShop. Different from previous work e.g. WoB, WebShop is more challenging with realistic linguistic elements. In order to better train an agent which can interact with WebShop, 1600 human trajectories are collected. Based on these human trajectories, reinforcement learning and imitation learning methods are adopted to train the agent. The experiments show that the trained agent performs better than a simple rule-based baseline, but far behind human performance.

**Questions:**

* Whether vision-language pre-trained models, e.g. CLIP, can benefit the proposed model?

**Limitations:**

The authors have pointed out some limitations about their work.

**Strengths And Weaknesses:**

(1) Strengths
* The paper is generally well written.
* Both simulation evaluation and sim-to-real transfer are conducted to validate the effectiveness of the proposed framework.

(2) Weaknesses
* It seems that WebShop focuses on search-based websites. I'm wondering whether it can be extended to more general websites. For example, whether the proposed actions (search, choose) are enough for general websites? If some website needs text input, how can we obtain these web pages for building simulation environments?
* Lack of some ablations in experiments. For example, both image and text information are used in the model. What's the effect if image Information is removed?

---

> ### Author Response · Authors · 2022-08-01
> **Thank you - Individual Response to smJp**
>
> 1. **Extension to more general websites, and whether the proposed actions (search, choose) are enough for general websites**
>
> This is a great point of discussion for future research.
>
> Compared with prior work, we believe WebShop is a significant step toward more general and scalable web task solving. For example, WoB uses synthetic webpages with a low-level action space such as moving mouse or typing keyboard keys. WikiNav only has an action space of link transitions among Wikipedia pages. Instead, we scrape large-scale real-world webpages, and the action space of searching text queries and choosing text buttons are general enough for most real-world e-commerce websites, as our sim2real transfer experiments (Section 5.3 in original draft) have shown.
>
> For future benchmarks with different websites, there could be extra actions involved. For example, in a flight website form filling (e.g. fill in destination text box) might be needed. From a modeling perspective, this is a composition of text selection (what box to fill in) and text generation (what to fill in) problems, so our model development for searching (text generation) and choosing (text selection) could still be useful. However, scraping a flight website could be more challenging than a shopping website and requires more fine-grained web actions during scraping, so some degree of database simulation (e.g. how price changes wrt date for different cities) might be needed. A bigger issue for extension to general websites could be task and reward formulation: for example, what task could be defined on a blog website? This is why we use e-commerce websites as a good sweet spot in this work: the data is rich, complex yet open-access, and a task can be well defined. We plan on extending this work to other domains and tasks and welcome more discussion/suggestions on future extensions.
>
> 2. **Image Ablation and whether pre-trained vision-language models can benefit**
>
> We perform an image ablation study as suggested (due to time limitation, we focus on the IL model), by removing the ResNet and the visual representation. We train 3 trials with different random seeds for both the IL model and the ablated IL model without images, with performances over 500 test cases below.
>
> | | score | success rate (\%) |
> | -------- | -------- | -------- |
> | IL  |   60.56 (1.94)  |  29 (2.42)   |
> | IL (w/o image)  |  60.33 (0.47)     | 28.4 (0.87)  |
>
> So ablating the image only slightly hurts the overall performance, but significantly reduces the variance. This is reasonable as our current instruction and reward setups only use textual information, and we believe future efforts to incorporate visual information into the task setup will better challenge models’ visual understanding, and make pre-trained vision-language models such as CLIP more useful. We expect these results to similarly hold for RL and RL+IL models as well, and will include these results in the camera ready.

---

> ### Author Response · Authors · 2022-08-08
> **Thank you again**
>
> Dear reviewer,
> Thank you again for your review -- please let us know if you have any remaining questions or concerns, so that we can address them before the deadline tomorrow. Alternatively, if you feel that your original concerns are addressed, we would appreciate your updating your evaluation to reflect that.

---

### Official Review · Reviewer_kKZ4 · 2022-07-11

**Rating:** 6
**Confidence:** 3
**Soundness:** 3 good
**Presentation:** 3 good
**Contribution:** 3 good

**Summary:**

The authors propose a new web interaction benchmark called WebShop, where the agent is requested to the navigate e-commerce web-page to buy an targeted items grounded with natural language formed instruction. To this end, they collect a number of real-world products, crowd-sourced text instructions, and human trajectories. They build a baseline using text instructions and human trajectories with imitating learning and reinforcement learning, and reveal that the proposed task has much room for improvement. Transfering sim-to-real minimal loss of generalibility inditates the potential usage of the proposed benchmark.

**Questions:**

Rather than aware of all attributes and having fixed preferences for the item, partial and incremental aware of attributes and having flexible preferences seem more realistic scenario. Several iterations of QA sessions for narrowing down the user preferences before purchasing, like in the task-oriented dialog (e.g. MultiWOZ), make the benchmark more realistic, and it would bring a great chance to the community.

**Limitations:**

Including ablation study on the effect of using image feature would be needed.

**Strengths And Weaknesses:**

[Strengths]
- Well written paper
- Tacking important research topic - web navigation
- Release of high quality benchmark

[Weaknesses]
- Instruction they construct may have somewhat unrealistic assumptions: 1) all attributes are known to be user before searching, 2) preferences are defined before searching the items, and 3) even those conditions are not changed during browsing the web shop.

---

> ### Author Response · Authors · 2022-08-01
> **Thank you - Individual Response to kKZ4**
>
> 1. **All attributes are known to be user before searching, preferences are defined before searching the items, even those conditions are not changed during browsing the web shop. Several iterations of QA sessions for narrowing down the user preferences before purchasing, like in the task-oriented dialog (e.g. MultiWOZ), make the benchmark more realistic**
>
> We agree that enabling user interaction in addition to web interaction will make agents more practical and useful for online shopping applications and think this is a great direction for future work. While many language tasks and datasets have been proposed based on user interaction (e.g. conservational QA, task-oriented dialogue, interactive semantic parsing), web interaction is a relatively under-explored direction where previous works (e.g. World of Bits) usually consider small-scale or synthetic tasks. So our work focuses on scaling up real-world web interaction, and demonstrates unique scalability advantages (i.e. web language data and transitions can be easily, quickly, and cheaply scraped in scale) and research challenges (e.g. query reformulation, stargatic exploration) compared to user interaction tasks. Our design of attribute-based instruction following and automatic reward function aims to support automatic and scalable reinforcement learning, which is hard for user interaction tasks due to the difficulty of human-in-the-loop training. So as a benchmark for interactive language agents, we believe WebShop is a good complement to previous user interaction tasks, and a good starting point for future research that incorporates user interaction into web-based tasks for more practical applications.
>
> 2. **Including ablation study on the effect of using image features would be needed.**
>
> We perform an image ablation study as suggested (due to time limitation, we focus on the IL model), by removing the ResNet and the visual representation. We train 3 trials with different random seeds for both the IL model and the ablated IL model without images, with performances over 500 test cases below.
>
> | | score | success rate (\%) |
> | -------- | -------- | -------- |
> | IL  |   60.56 (1.94)  |  29 (2.42)   |
> | IL (w/o image)  |  60.33 (0.47)     | 28.4 (0.87)  |
>
> So ablating the image only slightly hurts the overall performance, but significantly reduces the variance. This is reasonable as our current instruction and reward setups only use textual information, and we believe future efforts to incorporate visual information into the task setup will better challenge models’ visual understanding, and make pre-trained vision-language models such as CLIP more useful. We expect these results to similarly hold for RL and RL+IL models as well, and will include these results in the camera ready.

---

> ### Author Response · Authors · 2022-08-08
> **Thank you again**
>
> Dear reviewer,
> Thank you again for your review -- please let us know if you have any remaining questions or concerns, so that we can address them before the deadline tomorrow. Alternatively, if you feel that your original concerns are addressed, we would appreciate your updating your evaluation to reflect that.

---

### Author Response · Authors · 2022-08-01
**General Response and Draft Revision**

We want to thank all reviewers for their helpful feedback, which has been incorporated into our revised paper draft. Here we briefly reiterate the motivation and significance of our work.

**WebShop is an initial step towards using large-scale real-world web interaction as a research benchmark**, and we aim to demonstrate several unique benefits of this new direction for studying grounded language understanding and decision making. Compared to previous interactive language benchmarks (e.g. dialogue, conversational QA) where collecting human interaction is expensive and non-scalable, web interaction benchmarks like WebShop can leverage large amounts of realistic data (language and other modalities like vision) and transitions scraped from the Internet to support scalable interactive learning. Compared to previous reinforcement learning environments (e.g. video games, robotics), WebShop contains real-world language use and supports scaling of the simulated environment, on which trained agents could more easily transfer to useful real-world tasks, e.g. shopping directly on amazon.com.

We are excited about this direction and its many future extensions, e.g. incorporation of user interaction (kKZ4 #1), more use of multi-modal elements (kKZ4 #2, smJp #2), extension to more general websites (smJp #1), and we believe the disentanglement of the environment from the specific task in our paradigm is particularly helpful for these future research efforts. For example, with the same website environment we can improve the instructions and rewards to better use images, and we can also easily incorporate more web pages or elements (e.g. user review) to the website environment while keeping the task fixed. We can also easily extend the environment to contain new web-based domains and tasks. Progress on these benchmarks will also lead to practical use cases for language-grounded RL agents for solving tedious tasks for humans (section 5.4 in the updated draft) such as finding and comparing online shopping products and improving quality of human life in general (ytjW #4).

**Draft Revisions**: We have incorporated reviewer feedback and supplementary texts to strengthen the claims and findings into our revised draft (text highlighted in blue) including
1) ablation on image in section C.5 (kKZ4 & smJP),
2) ablation on beam search vs. top-1 in section C.4 (ytjW)
3) clarifying edits in the main text in L268 (ytjW)
4) additional sim2real experiments on eBay.com in section 5.4
5) detailed analysis on human trajectory collection in section A.6 (ytjW)
6) faithfulness of our automatic reward compared to human inspection in section A.7 (ytjW)

---

### Author Response · Authors · 2022-08-07
**Look forward to discussion**

Dear AC and reviewers:

Thank all reviewers for the constructive comments, which have helped us significantly improve the draft!

Since the discussion period is about to end and we have not heard any post-rebuttal response yet, please don’t hesitate to let us know if there is any additional information that we can offer, as we would love to convince you of the merits of the paper and the direction of large-scale real-world web interaction as a research problem. Thanks again!

---

> ### Author Response · Authors · 2022-08-09
> **A summary of rebuttal and discussion at the attention of the AC (and the reviewers)**
>
> We want to thank reviewers again for their time and valuable questions. As the discussion period is about to end and we don't get to hear back from all reviewers yet, we'd like to briefly summarize our rebuttal and updates:
>
> - We reiterated the motivation of our work, and why we believe large-scale real-world web interaction is an exciting direction to build research benchmarks for language grounding and decision making.
> - We addressed all experimental (e.g. image ablation, beam search) and conceptual (e.g. user interaction, more general actions and websites, human performance) questions of reviewers with new experiments and discussions incorporated into the paper.
> - We also updated the paper with additional results (e.g. sim2real transfer to ebay) to make the result stronger.

---

### Meta-Review · Area_Chair_Daht · 2022-08-26

**Recommendation:** Accept
**Confidence:** Less certain

**Metareview:**

This paper proposes a new real-world natural language web interaction benchmark for the shopping domain and associated dataset, WebShop. The paper includes a rule-based baseline, as well as a model-based one that is trained using imitation and reinforcement learning. Performance is reported using these models in addition to human performance, showing a large gap and opportunity for better models. Reviewers made a good set of suggestions, majority of which have been covered by the authors in their rebuttal. These include additional experimentation requested by the reviewers as well as improvements to the paper presentation.

**Award:**

No

---

### Decision · Program_Chairs · 2022-09-14

Accept